# Phenotypic Analysis and Gene Cloning of Rice Floury Endosperm Mutant *wcr* (White-Core Rice)

**DOI:** 10.3390/plants13182653

**Published:** 2024-09-22

**Authors:** Yihao Yang, Xiaoyi Yang, Lingjun Wu, Zixing Sun, Yi Zhang, Ziyan Shen, Juan Zhou, Min Guo, Changjie Yan

**Affiliations:** 1Jiangsu Key Laboratory of Crop Genomics and Molecular Breeding/Zhongshan Biological Breeding Laboratory/Key Laboratory of Plant Functional Genomics of the Ministry of Education, Agricultural College, Yangzhou University, Yangzhou 225009, China; 2Jiangsu Co-Innovation Center for Modern Production Technology of Grain Crops/Jiangsu Key Laboratory of Crop Genetics and Physiology, Yangzhou University, Yangzhou 225009, China

**Keywords:** grain quality, storage substances, *OsPPDKB*

## Abstract

The composition and distribution of storage substances in rice endosperm directly affect grain quality. A floury endosperm mutant, *wcr* (white-core rice), was identified, exhibiting a loose arrangement of starch granules with a floury opaque appearance in the inner layer of mature grains, resulting in reduced grain weight. The total starch and amylose content remained unchanged, but the levels of the four component proteins in the mutant brown rice significantly decreased. Additionally, the milled rice (inner endosperm) showed a significant decrease in total starch and amylose content, accompanied by a nearly threefold increase in albumin content. The swelling capacity of mutant starch was reduced, and its chain length distribution was altered. The target gene was mapped on chromosome 5 within a 65 kb region. A frameshift mutation occurred due to an insertion of an extra C base in the second exon of the *cyOsPPDKB* gene, which encodes pyruvate phosphate dikinase. Expression analysis revealed that *wcr* not only affected genes involved in starch metabolism but also downregulated expression levels of genes associated with storage protein synthesis. Overall, *wcr* plays a crucial role as a regulator factor influencing protein synthesis and starch metabolism in rice grains.

## 1. Introduction

Rice (*Oryza sativa* L.) is a crucial staple crop worldwide, and the advancements in dwarf breeding during the 1960s, as well as the theoretical and technical breakthroughs in hybrid rice breeding during the 1970s, have significantly improved rice production in China, effectively addressing food scarcity issues [1]. With an improvement in living standards, there has been an increasing demand among consumers for higher rice grain quality. Starch and protein are the primary storage substances within the endosperm of rice grains. Starch constitutes a majority portion that determines both grain quality and yield, followed by protein content which influences rice palatability and nutritional value [2,3]. Therefore, it is of immense significance to comprehensively analyze regulatory mechanisms governing starch and protein synthesis as well as accumulation while exploring associated genes to enhance rice grain quality.

The rice floury endosperm mutants are valuable genetic resources for studying the complex networks involved in endosperm development and quality regulation due to their abnormal synthesis and accumulation of starch or storage proteins, resulting in floury and opaque endosperms. Currently, a multitude of genes associated with floury endosperm mutants have been successfully cloned, encompassing diverse aspects of cellular metabolic processes including starch synthesis and sucrose metabolism, amyloplastic development, energy supply, glycolytic metabolism, protein processing and transport, lipid transport, epigenetics, transcriptional regulation, and protein interaction (Table 1). The cloning and molecular mechanism analysis of the aforementioned genes provide a theoretical foundation for understanding starch and protein synthesis, as well as the biochemical metabolic pathways in rice. This enhances our knowledge of endosperm development regulation and offers the potential for improving rice quality through genetic engineering.

The synthesis and transportation of storage substances in cereal grain endosperm require sufficient energy supply [53]. However, the low oxygen environment within the endosperm hampers oxidative phosphorylation, thereby limiting ATP production [54,55]. The glycolytic pathway serves as a crucial means for biological organisms to generate energy under anaerobic conditions, with pyruvate kinase (PK) and pyruvate phosphate dikinase (PPDK) catalyzing this reaction [56,57]. In plants, PPDK can be classified into chloroplast chPPDK and cytosolic cyPPDK. ChPPDK is primarily found in chloroplasts and expressed predominantly in photosynthetic tissues such as leaves of C4 plants. Conversely, cyPPDK is mainly localized within the cytoplasm and expressed in non-photosynthetic tissues like grains and roots [58]. In rice, there are only two genes encoding PPDK: one encodes OsPPDKA while another produces two transcripts of C4-type chloroplastic chOsPPDKB and cytosolic cyOsPPDKB [24]. CyOsPPDKB is responsible for the reversible conversion of pyruvate and inorganic phosphate (Pi) into phosphoenolpyruvate (PEP) and inorganic pyrophosphate (PPi), which can serve as an alternative phosphate donor for ATP in plant cells [24,59,60,61]. This process provides essential carbon skeletons for amino acid and lipid biosynthesis, directly influencing ADP-glucose allocation towards starch and lipids [24,62,63,64,65,66,67,68].

In this study, we identified a white-core endosperm mutant (*wcr*) derived from the rice variety Sasanishiki through tissue culture. We present a comprehensive investigation of the starch morphology, physicochemical properties, and protein composition content of the grains. Additionally, we precisely mapped the mutant gene and explored its involvement in regulating the network that governs rice grain starch and storage protein content. This research contributes valuable genetic resources and provides a theoretical foundation for unraveling mechanisms underlying rice endosperm development and grain quality regulation.

## 2. Results

### 2.1. Analysis of Phenotypic and Crop Traits of the wcr Mutant

The *wcr* mutant was derived from the *Japonica* rice variety Sasanishiki through plant tissue culture. Overall, there were minimal changes in the plant architecture of the *wcr* mutant (Figure 1A). Upon hull removal, the mature grain of the *wcr* mutant exhibits a starchy and translucent state, with opaqueness primarily localized in the inner endosperm (Figure 1B,C). The starch granules in the endosperm of wild-type exhibited a tightly and evenly arranged pattern in both inner and outer layers under electron microscope scanning (Figure 1D). In contrast, the starch granules in *wcr*’s endosperm showed a loosely arranged structure with larger inter-granular gaps and an increased presence of single-grain type starch granules within the inner layer (Figure 1D). The mutant showed significantly increased plant height, grain length, and width compared to the wild type (Figure 1A,E,G,H), while exhibiting a significant decrease in tiller number, grain thickness, and thousand-grain weight (Figure 1F,I,J).

### 2.2. Physicochemical Characteristics of Mature Grains of wcr Mutant

The physicochemical properties of brown rice and milled rice from the *wcr* mutant were further investigated in comparison to those of the wild type, due to its abnormal endosperm development and significant differences observed in starch granule structure between the inner and outer layers (Figure 1B–D). No significant differences were observed in the total starch content and amylose content of brown rice flour between *wcr* and wild type (Figure 2A,B). However, both traits showed a significant decrease when analyzing milled rice flour (Figure 2A,B). Based on these results, the total starch content and amylose content in the peripheral endosperm of *wcr* significantly increased compared to the wild type (Figure 2H). The analysis of storage proteins showed a significant decrease in all four component proteins (albumin, globulin, prolamin, and glutelin) in mutant brown rice flour (Figure 2C–F). However, apart from a nearly threefold increase in the content of albumin compared to the wild type (Figure 2C), there is no difference in the protein content of the other three components when analyzed in milled rice flour (Figure 2D–F). These changes were also reflected in the SDS-PAGE analysis of storage proteins (Figure 2G). It can be inferred that unlike the changing trends of total starch content and amylose content in the *wcr* peripheral endosperm, protein contents for all four components significantly decrease compared to the wild-type with albumin showing the largest decrease (Figure 2H).

### 2.3. Gelatinization Characteristics and Amylopectin Structure Analysis of wcr

Although the total starch content and amylose content in the brown rice flour of the mutant did not show significant differences compared to those of the wild type (Figure 2A,B), results from starch gelatinization tests using varying concentrations of urea solution revealed that the *wcr* mutant exhibited reduced solubility in urea. As shown in Figure 3A,B, the swelling volume of the mutant’s brown rice flour was smaller than that of the wild-type under different urea concentrations, with a noticeable difference observed at a concentration of 1 mol/L (Figure 3B). At a urea concentration of 8 mol/L, no expansion was observed for the mutant while it continued for the wild type (Figure 3A,B). Furthermore, the analysis of the amylopectin chain length distribution revealed a reduction in the content of chain lengths ranging from 6 to 35 in the degree of polymerization (DP) in the brown rice flour of the *wcr* mutant, while an increase was observed in the chain lengths greater than 36 DP (Figure 3C).

### 2.4. Genetic Analysis of the wcr Mutant

The *wcr* mutant was crossed with the background Sasanishiki, resulting in F_1_ plants with normal grain development. In the subsequent F_2_ population, segregation occurred between plants with normal endosperm and those with floury endosperm. A random survey of 232 F_2_ plants showed that 177 had normal phenotypes, while 55 had mutant phenotypes. The observed segregation pattern of 3:1 (*x*^2^ = 0.143 < 3.84) in the F_2_ population suggests that the *wcr* mutant is governed by a pair of recessive nuclear genes.

### 2.5. Fine Mapping of the wcr Gene

We successfully mapped the *wcr* gene to chromosome 5 at approximately 5.41 Mb (19,268,954 bp–24,757,870 bp) physical interval using MutMap analysis (Figure 4A). To further fine-map the *wcr* gene, we crossed the *wcr* mutant with the single-segment substitution line SL418 [69], resulting in an F_2_ population of 2074 plants. From this population, we selected 108 plants with floury endosperm and narrowed down the location of the target gene within a 76 kb region delimited by markers C5.8 and C5.9 (Table 2, Figure 4B), which encompasses eight *ORFs*. Among these *ORFs*, *LOC_Os05g33570* encoding a functional pyruvate phosphate dikinase *OsPPDKB* was found to be allelic to the *flo4* gene [24] (Figure 4C). Sequencing results revealed a frameshift mutation caused by an insertion of an additional base C into the second exon of *cyOsPPDKB* (Figure 4D). Therefore, *wcr* represents a novel allelic variation of *flo4*.

### 2.6. Gene Expression Analysis of Storage Substance-Related Genes

Considering the alterations in storage substances (starch and protein) composition and distribution in the *wcr* mutant (Figure 2), we conducted a comprehensive investigation into the expression levels of 35 genes involved in grain protein synthesis and 18 genes associated with starch metabolism in both wild type and mutant rice. As shown in Figure 5, the expression levels of most rice protein synthesis genes in the mutant were significantly reduced (Figure 5A), while the expression levels of starch metabolism-related genes exhibited varying degrees of downregulation, but the overall magnitude was small (Figure 5B). 

## 3. Discussion

Through gene mapping and sequencing comparison, it has been determined that the *wcr* mutant is governed by a pair of recessive nuclear genes encoding pyruvate phosphate dikinase *OsPPDKB*. A comprehensive literature search has identified a total of nine distinct allelic mutant genes for *OsPPDKB*, including knockout mutations (*flo4-1* [24], *flo4-2* [24], *flo4-3* [24], and *flo4-303* [63]) and missense mutations (*flo4-4* [64], *flo4-5* [65], *flo4-6* [66], *M14* [67], and *floTR1* [68]) (Figure 6). The *wcr* mutation in this study was caused by an additional C base insertion in the second exon of *cyOsPPDKB*, resulting in a frameshift mutation. Therefore, the *wcr* can be considered as a novel allelic gene variant of *cyOsPPDKB*. Although there have been some changes in the crop traits of *wcr,* such as increased plant height and reduced tillering, which differ from previous studies, it should be noted that the *wcr* mutant originated from plant tissue culture, where stable heritable variations were likely generated during the process.

The 10 identified mutants exhibit a similar floury endosperm phenotype but show significant variation in the physicochemical properties of rice flour, including amylose content, protein content, and amylopectin structure. In terms of amylose content, most allelic mutant grains show a significant reduction ranging from 13% to 46%, with *flo4-4* showing a 5% significant increase and *flo4-5* having no impact on amylose levels [24,63,64,65,66,67,68]. Interestingly, while the total starch and amylose content in the brown rice of the *wcr* mutant was similar to those of the wild type, there was a significant decrease observed in milled rice (Figure 2A,B). Regarding total protein content, *flo4-1*, *flo4-2,* and *floTR1* mutant grains showed an increase ranging from 5% to 16%, while *flo4-4* and *flo4-5* alleles resulted in a decrease of more than 15% [24,65]. No effect on protein was observed for *M14* rice grains [67]. Notably, the levels of four component proteins were significantly reduced in the *wcr* mutant brown rice; however, apart from a notable increase in albumin content observed for milled rice (Figure 2C–F), minimal changes were seen for the other three component proteins. The analysis of amylopectin structure revealed that both *floTR1* and *M14* grains contained fewer short chains (DP < 16) but more middle-length chains (DP > 18) [67,68], which diverged from our study’s findings. In contrast, the *wcr* mutant showed fewer middle-length chains (DP < 36) but more long chains (DP > 36) (Figure 3C). In summary, different allelic mutations of *cyOsPPDKB* resulted in significant changes in the content and distribution of rice storage substances across various genetic backgrounds; however, these changes were not entirely consistent. We propose that the diverse physicochemical properties observed among different allelic mutants of *cyOsPPDKB* may be attributed to several potential factors. Firstly, distinct allelic variants of *cyOsPPDKB* may generate varying protein functionalities; however, further confirmation through molecular genetic experiments is required for validation. Secondly, dissimilar genetic backgrounds among the mutants are likely to contribute to the observed differences in physicochemical properties, which could be influenced by combined effects with other genes within the genome. Therefore, future investigations can explore utilizing CRISPR/Cas9 technology to generate saturated mutations of *cyOsPPDKB* within a consistent genetic background and fully elucidate its molecular functions.

The distinguishing feature of the *OsPPDK* mutant, compared to other mutants with rice floury endosperm, is its translucent periphery and internal flouriness. In this study, the arrangement of amyloplasts in the inner endosperm and peripheral regions of *wcr* grains was observed using scanning electron microscopy. It was found that internal amyloplasts had a loose structure with larger pores, while external ones were densely packed (Figure 1D). The reasons for this phenomenon may be as follows: (1) Previous research has shown that OsPPDKB plays a compensatory role in ATP deficiency under anaerobic conditions [24,57,58,59,60,61]. In normal rice grains, the level of hypoxia is more severe in the inner endosperm than in the peripheral region. When there is a mutation in *OsPPDKB* and it loses its ability to convert pyruvate and inorganic phosphate (PI) into phosphoenolpyruvate (PEP) and inorganic pyrophosphate (PPI), which can serve as an alternative phosphate donor for ATP production [24,57,58,59,60,61], substance transport within the inner endosperm is disrupted, leading to poor endosperm development. However, there is an improvement in aerobic conditions within the grain peripheral region, likely providing sufficient energy for material transport through other oxidative phosphorylation pathways, resulting in relatively normal development. (2) The expression of *cyOsPPDKB* in rice grains peaks at 10 days after flowering, followed by a rapid decrease in protein level and activity through threonyl-phosphorylation and protein degradation mechanisms around approximately 20 days after anthesis [61]. This suggests that *cyOsPPDKB* likely plays a crucial role during the initial 20-day period of grain filling, which coincides with the accumulation of storage substances from the central region to the periphery of the endosperm [70]. Therefore, *cyOsPPDKB*’s temporal and spatial expression pattern is likely associated with severe flouriness in the inner endosperm and light flouriness in the outer part. However, these reasons are speculative hypotheses and require rigorous scientific experiments for validation.

With the diversification in dietary habits and rapid economic development, rice-processing products like rice cakes, noodles, crackers, and wine have become increasingly popular. Compared to regular rice cultivars, floury mutants with loosely packed starch granules are particularly useful for dry-milled flour production or sake brewing due to their easily breakable soft endosperm, finer particle size and less damaged starch, and strong water absorption properties [71]. For example, South Korea has developed new floury rice varieties such as suweon542 [72], Hangaru [73], Shingil [74], and Garumi2 [71] in recent years, which successfully reduced the rice milling cost. In this study, the starch granules of *wcr* endosperm were also loosely arranged, providing new genetic material for breeding low milling cost rice material. Additionally, previous research has shown that the spatial distribution and characteristics of protein content and composition in *Japonica* rice grains play a crucial role in determining the taste quality of sake [75]. In this study, the total protein content and component protein content in the *wcr* mutant brown rice were significantly reduced, accompanied by alterations in the distribution of different components. These findings highlight its potential as a valuable germplasm resource for further research aimed at enhancing sake quality.

## 4. Materials and Methods

### 4.1. Experimental Materials and Field Design

The experimental materials utilized in this study included a rice endosperm mutant *wcr* obtained through plant tissue culture, the parental variety Sasanishiki as the background control, an F_2_ segregating population derived from crossing *wcr* with Sasanishiki for MutMap analysis, and an F_2_ population obtained by crossing *wcr* with single-segment substitution line SL418 [69] for precise gene mapping. Each genotype was individually planted with 10 plants per row. The spacing between plants was maintained at 20 cm × 20 cm. Standardized water and fertilizer management practices were followed.

### 4.2. Main Crop Characteristics of Rice Plants

The main panicles of 5 mutants and 5 wild-type rice plants were selected at the mature stage for the investigation of height, tiller, grain length, grain width, and 1000-grain weight. The grain length and width of 5 mutants and 5 wild-type rice grains were analyzed using a seed-measuring instrument (Model SC-G, Wanshen, Hangzhou, China).

### 4.3. Scanning Electron Microscopy Analysis

The mature grains were transversely cut by a knife, coated with gold, and examined under a scanning electron microscope (SEM, S-4800, Hitachi, Tokyo, Japan). For the observation of compound starch granules, transverse sections of WT and *wcr* mature grains were fixed overnight in 2% (*v*/*v*) glutaraldehyde (CAS: P1126, Solarbio, Beijing, China). The samples were dehydrated in an alcohol series, embedded in LR White resin (Heraeus Kulzer, Wehrheim, Germany), and sectioned using an ultrathin microtome (EM UC7, Leica, Wetzlar, Germany). Semi-thin sections were dried at 40 °C, stained with I2-KI solution for visualization purposes, and observed using a light microscope (BX53 Olympus, Tokyo, Japan).

### 4.4. Protein Extraction and SDS-PAGE Analysis

The four kinds of storage protein were extracted from the floury grains of 3 mutants and 3 wild-type rice grains as described previously [76], and the protein content of each component was quantified by means of the Bradford assay [77]. SDS-PAGE and protein gel blot analysis were performed as described previously [78].

### 4.5. Analysis of the Amylose and Total Starch Contents

The total starch content of 3 mutants and 3 wild-type rice grains was measured using a Megazyme Inc. kit (Bray, Ireland) following the procedure provided by the manufacturer. The apparent amylose content of 3 mutants and 3 wild-type rice grains was measured using the iodine colorimetric method [79].

### 4.6. Determination of Chain Length of Amylopectin

Starch (10 mg) was dissolved in 5 mL water in a water bath (100 °C) for 60 min; 10 microliters (μL) of sodium azide solution (2% *w*/*v*) (CAS: 26628-22-8, Anpel, Shanghai, China), 50 μL acetate buffer (0.6 M, pH 4.4) (CAS: 127-09-3, Anpel, China), and 10 μL isoamylase (1400 U) (08124, Sigma, Darmstadt, Germany) were added to the starch dispersion, and the mixture was incubated in a water bath at 37 °C for 24 h. The hydroxyl groups of the debranched glucans were reduced by treatment with 0.5% (*w*/*v*) of sodium borohydride (16940-66-2, Sigma, Germany) under alkaline conditions for 20 h. The preparation of about 600 μL was dried in a vacuum at room temperature and allowed to dissolve in 30 μL of 1 M NaOH (16940-66-2, Anpel, China) for 60 min. Then, the solution was diluted with 570 μL of distilled water.

The sample extracts were analyzed by high-performance anion-exchange chromatography (HPAEC) on a CarboPac PA-200 anion-exchange column (4.0 × 250 mm; Dionex, Sunnyvale, CA, USA) using a pulsed amperometric detector (PAD; Dionex ICS 5000 system): flow rate, 0.4 mL/min; injection volume, 5 μL; solvent system, 0.2 M NaOH: (0.2 M NaOH, 0.2 M NaAc); gradient program, 90:10 *v*/*v* at 0 min, 90:10 *v/v* at 10 min, 40:60 *v/v* at 30 min, 40:60 *v/v* at 50 min, 90:10 *v/v* at 50.1 min, 90:10 *v/v* at 60 min.

Data were acquired on the ICS5000 (Thermo Scientific, Waltham, MA, USA) and processed using chromeleon 7.2 CDS (Thermo Scientific). Quantified data were output into Excel format.

### 4.7. Genome Mapping of the wcr Locus

To clone the *wcr* gene, the *wcr* mutant was used as the parent for a cross with the background Sasanishiki to produce an F_2_ population. In the F_2_ population, we selected 20 plants that showed *wcr* phenotypes and pooled their DNA in an equal ratio for whole-genome resequencing using NextSeq 500 (Illumina, San Diego, CA, USA, http://www.illumina.com/). The MutMap was performed according to a previous study [80]. To further fine-mapping the *wcr* gene, we crossed the *wcr* mutant with the single-segment substitution line SL418 to create an F_2_ population of 2074 plants. By utilizing nine polymorphic markers, we successfully fine-mapped the target gene. The primers are listed in Table 2.

### 4.8. RNA Extraction, cDNA Preparation, and qRT-PCR

Total RNA was extracted from endosperm 15 days after flowering using an RNA extraction kit (Tiangen, Beijing, China). First-strand cDNA was synthesized using a reverse transcription kit (Vazyme, Nanjing, China). Quantitative reverse transcription (qRT-PCR) was performed with a CFX96 Real-Time PCR System (Bio-Rad, Hercules, CA, USA) using an SYBR qPCR Master Mix (Vazyme, Nanjing, China). The PCR procedure was carried out using the following program: 95 °C for 10 min, then 40 cycles of 95 °C for 15 s, and 60 °C for 1 min. All assays were performed with at least three biological replicates; each biological study repeats the setup three times for technical replication. The rice *actin* gene served as the internal control to normalize gene expression. Quantitative gene expression was analyzed from three biological replicates by the 2^−ΔΔCt^ method. The primers were used as described previously [76].

### 4.9. Data Analysis

The experiment was conducted with three biological replicates for each sample. Microsoft Excel 2016 was used for data collection, while the statistical analysis software SPSS 15.0 was used for variance analysis. All experimental data are presented as means ± SD.

## Figures and Tables

**Figure 1 plants-13-02653-f001:**
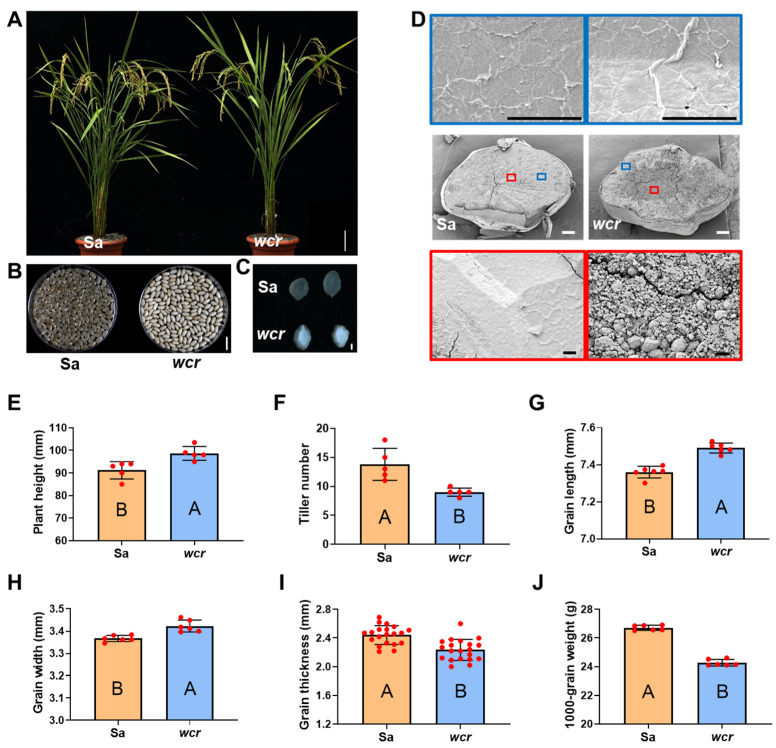
Phenotypic and crop traits analysis of the *wcr* mutant and wild type. (**A**) The whole plants of the Sa-Sasanishiki wild-type and *wcr*, scale bar = 10 cm. (**B**) The brown rice of the Sa and *wcr*, scale bar = 10 mm. (**C**) The cross-section of the Sa and *wcr* grains, scale bar = 1 mm. (**D**) Electron microscopic scanning of the endosperm of Sa and *wcr*, black scale bar = 10 μm, white scale bar = 200 μm. Different colored boxes represent local enlargements of starch structures. (**E**–**J**) The crop traits of Sa and *wcr*. Different upper case letters denote significant statistical differences between Sa (orange) and *wcr* (blue) plants, with the *p*-value < 0.01.

**Figure 2 plants-13-02653-f002:**
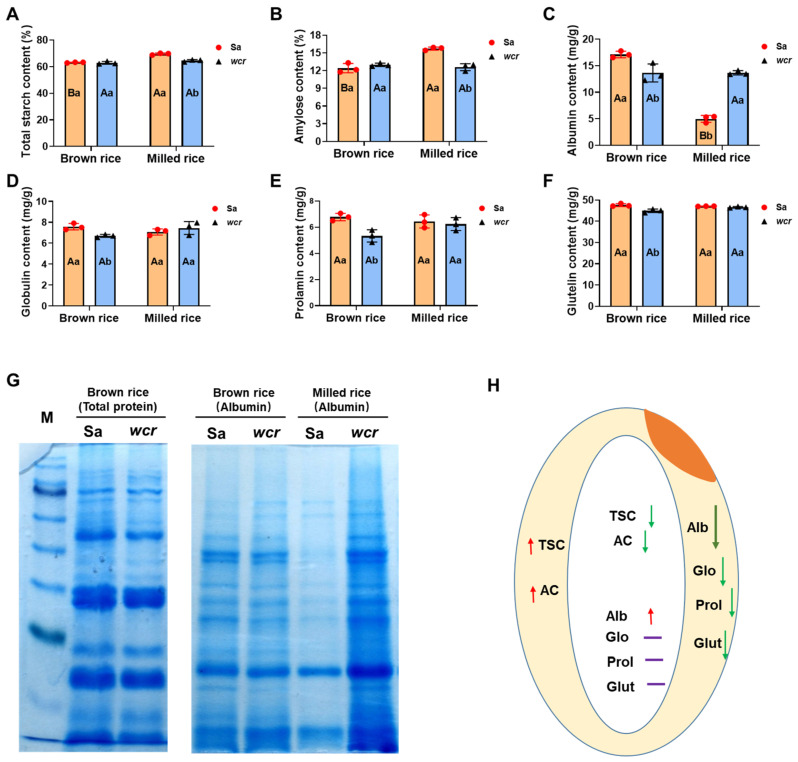
Physicochemical properties of mature grains of Sa and *wcr* mutant. (**A**–**F**) The contents of total starch, amylose, albumin, globulin, prolamin, and glutelin in brown rice flour and milled rice flour of Sa and *wcr* mutant. Different small case letters denote statistical differences between Sa (orange) and *wcr* (blue) plants in the same rice type (brown or milled); different upper case letters denote statistical differences between brown or milled rice types in the same wild type or *wcr* plants. (**G**) The SDS-PAGE analysis of storage proteins of Sa and *wcr* mutant rice flour. (**H**) Changes in the contents of storage substances in the inner and outer endosperm of *wcr* compared to wild type (the yellow part indicates the outer endosperm; the white part indicates the inner endosperm; Red arrows represent up-regulated levels, light green and dark green arrows represent down-regulated levels, and purple horizontal lines indicate unchanged levels; TSC, total starch content; AC, amylose content; Alb, albumin; Glo, globulin; Prol, prolamin; Glut, glutelin).

**Figure 3 plants-13-02653-f003:**
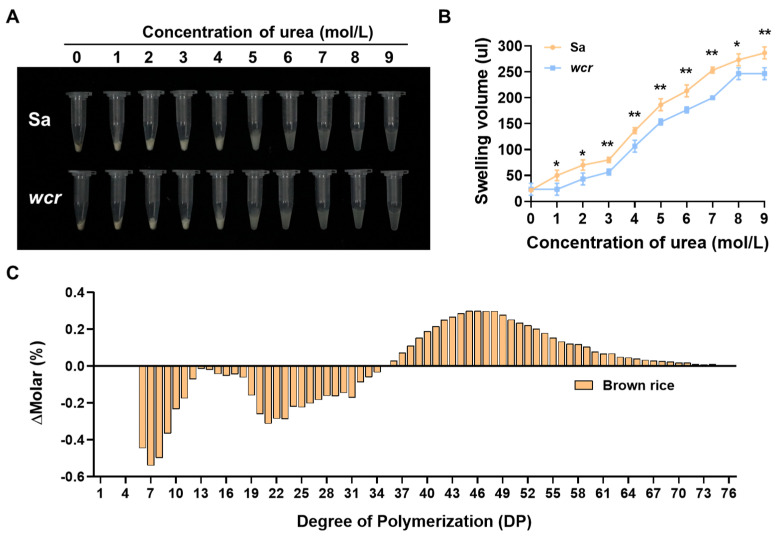
Gelatinization characteristics and amylopectin structure analysis of *wcr* mutant. (**A**,**B**) Comparison of the swelling volume of brown rice flour between Sa and *wcr* mutant under different urea concentrations. The *p*-values < 0.05 * and <0.01 ** calculated using an independent-samples *t*-test. (**C**) Determination of amylopectin chain length distribution in Sa and *wcr* mutant brown rice flour.

**Figure 4 plants-13-02653-f004:**
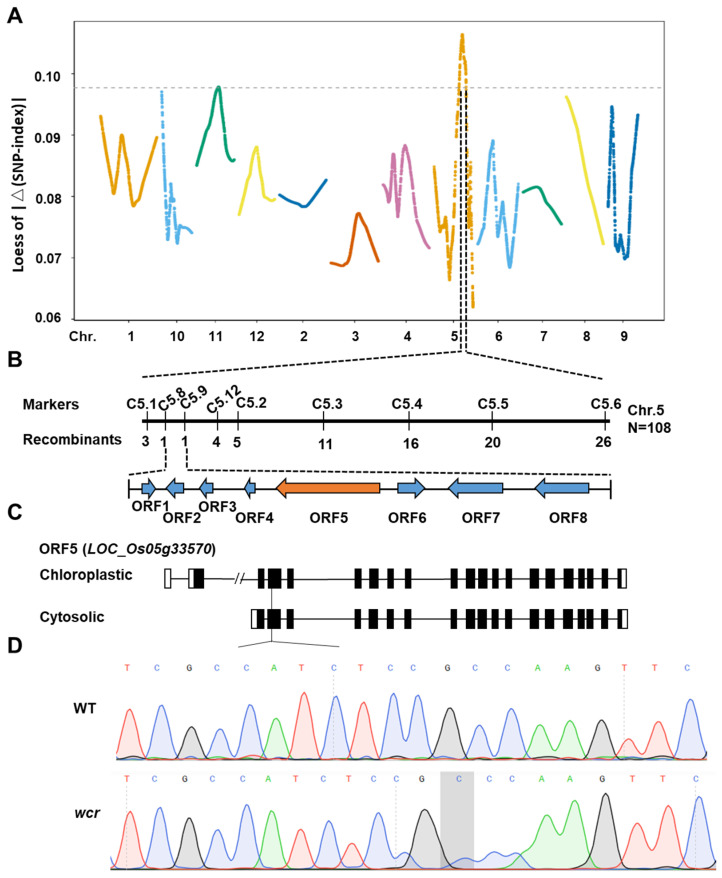
Fine mapping of the *wcr* gene. (**A**) The MutMap analysis of the *wcr* gene. (**B**) Fine mapping of the *wcr* gene using linkage analysis. (**C**) The gene structure of the *LOC_Os05g33570*. (**D**) The Sanger chromatogram of the WT and *wcr*.

**Figure 5 plants-13-02653-f005:**
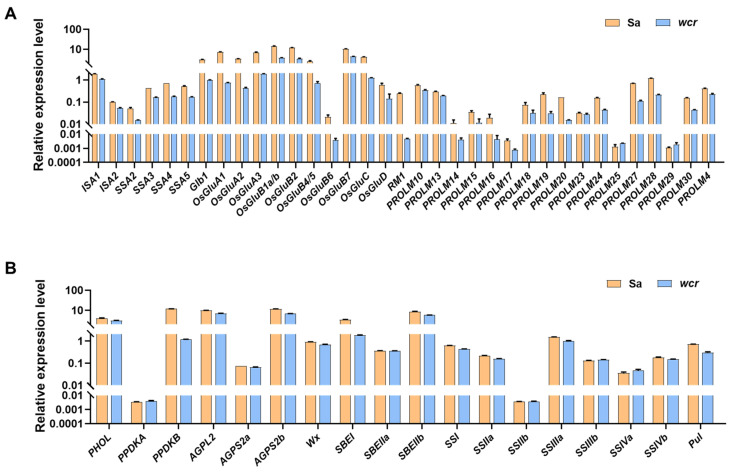
Gene expression analysis of storage substance-related genes of Sa and the *wcr* mutant. (**A**) Gene expression analysis of grain protein biosynthesis genes. (**B**) Gene expression analysis of grain starch metabolism genes.

**Figure 6 plants-13-02653-f006:**
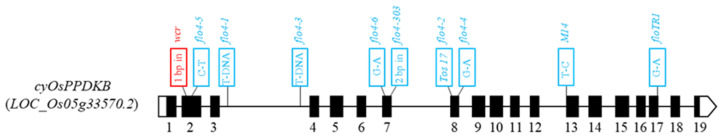
The structure of *cyOsPPDKB*. The *wcr* mutant is highlighted in red boxes. Previously reported allelic mutants of *cyOsPPDKB* are indicated in blue boxes.

**Table 1 plants-13-02653-t001:** Reported floury endosperm genes.

Classification	Name	Gene ID	Annotation
Starch synthesis and sucrose metabolism	*FLO8*	*Os09g0553200*	UTP-glucose-1-phosphate uridylyltransferase [4]
	*OsAGPL2*	*Os01g0633100*	Glucose-1-phosphate adenylyltransferase large subunit [5]
	*OsAGPS*	*Os09g0298200*	Glucose-1-phosphate adenylyltransferase large subunit [5]
	*OsBT1*	*Os02g0202400*	ADP-Glucose Transporter [6]
	*OsBEIIb*	*Os02g0528200*	1,4-alpha-glucan-branching enzyme [7]
	*FLO5*	*Os08g0191433*	Starch synthase III [8]
	*Pho1*	*Os03g0758100*	Alpha-glucan phosphorylase isozyme [9]
	*GIF1*	*Os04g0413500*	Cell wall invertase [10]
	*Wx*	*Os06g0133000*	Starch synthase [11]
	*FLO24*	*Os03g0426900*	Heat shock protein 101 [12]
Amyloplast development	*SSG4*	*Os01g0179400*	Protein containing a DUF490 domain [13]
	*SSG6*	*Os06g0130400*	Aminotransferase [14]
	*SSG7*	*Os11g0524300*	Plant-specific DUF1001 domain-containing protein [15]
	*FSE1*	*Os08g0110700*	Phospholipase-like protein [16]
Energy supply	*FLO13*	*Os02g0816800*	Mitochondrial complex I subunit [17]
	*OGR1*	*Os12g0270200*	Pentatricopeptide repeat–DYW protein [18]
	*FLO10*	*Os03g0168400*	Pentatricopeptide repeat protein [19]
	*OsNPPR1*	*Os08g0290000*	Pentatricopeptide repeat protein [20]
	*FLO18*	*Os07g0688100*	Pentatricopeptide repeat protein [21]
	*FLO22*	*Os07g0179000*	P-type pentatricopeptide repeat (PPR) protein [22]
Glycolytic metabolism	*PFPβ*	*Os06g0247500*	Pyrophosphate-fructose 6-phosphate 1-phosphotransferase [23]
	*FLO4*	*Os05g0405000*	Pyruvate, phosphate dikinase [24]
	*OsPK2*	*Os07g0181000*	Plastidic pyruvate kinase [25]
	*FLO12*	*Os10g0390500*	Aminotransferase [26]
	*FLO15*	*Os05g0230900*	Glyoxalase family protein [27]
	*FLO16*	*Os10g0478200*	Lactate/malate dehydrogenase [28]
	*FLO23*	*Os03g0294200*	Fructose-6-phosphate-2-kinase/fructose-2, 6-bisphosphatase [29]
	*FLO19*	*Os04g0119400*	Pyruvate dehydrogenase complex E1 component subunit α1 [30]
	*FLO19*	*Os03g0685300*	Class I glutamine amidotransferase [31]
Protein processing and transport	*PDIL1-1*	*Os11g0199200*	Protein disulfide isomerase-like enzyme [32]
	*GPA1*	*Os12g0631100*	Small GTPase [33]
	*GPA2*	*Os03g0262900*	Guanine nucleotide exchange factor [34]
	*GPA3*	*Os03g0835800*	Regulator of post-Golgi vesicular traffic [35]
	*GPA4*	*Os03g0209400*	Golgi Transport 1 [36]
	*GPA5*	*Os06g0643000*	Rab5a Effector [37]
	*GPA6*	*Os09g0286400*	Vacuolar Na^+^/H^+^ antiporter [38]
	*GPA7*	*Os08g0427300*	Homolog of Arabidopsis CCZ1a and CCZ1b [39]
	*GPA8*	*Os01g0659200*	Subunit E isoform 1 of vacuolar H^+^-ATPase [40]
Lipid transport	*ESG1*	*Os04g0553000*	Bacterial MlaE lipid transfer protein [41]
Epigenetics	*OsROS1*	*Os01g0218032*	DNA demethylase [42]
	*FLO20-1*	*Os01g0874900*	Serine hydroxymethyltransferase [43]
Transcriptional regulation and protein interaction	*RISBZ1*	*Os07g0182000*	bZIP transcription factor [44]
	*RSR1*	*Os05g0121600*	Transcription factor of the AP2/EREBP family [45]
	*NF-YB1*	*Os02g0725900*	Nuclear transcription factor Y subunit B [46]
	*NF-YC12*	*Os05g0304800*	CCAAT-box-binding transcription factor [46,47]
	*bHLH144*	*Os04g0429400*	Helix-loop-helix DNA-binding domain containing protein [46]
	*FLO2*	*Os04g0645100*	Tetratricopeptide repeat domain-containing protein [48]
	*FLO6*	*Os03g0686900*	CBM48 domain-containing protein [49]
	*FLO7*	*Os10g0463800*	DUF1388 domain protein [50]
	*FLO11*	*Os12g0244100*	Plastid heat shock protein 70 [51]
	*FLO9*	*Os11g0586300*	Homologous to Arabidopsis LIKE EARLY STARVATION1 [52]

**Table 2 plants-13-02653-t002:** The primers used for fine-mapping the *wcr.*

Name	Forward Primer (5′-3′)	Reverse Primer (5′-3′)
C5.5	CTATGCAGTGCAGTGTGCAC	AGCCGAAGGAGGTGTGAATC
C5.4	GCTCAAGCAAGGTCCATTCC	CAGCTACTAGGCCCCATTTG
C5.3	CCTGGCGTCAAACACATCTG	CTGAGGGTGTTCTTTTGGGC
C5.2	ATGGGAGAAGTGTCCAGCAG	GTGTGGACTGTGGATTGTGG
C5.1	AGAACGGAGGGAGTAGGATC	TCGCGGCTCTGAATTACCAG
C5.8	GTCCACCCGTTTCTTGCATG	CCACCCGTTTCTTGCATACC
C5.9	CCGGATTGTAGCTGTAGCTC	GGGTCACAGCATCAAAGCAG
C5.12	GTGCTGGAAACTCCATGTCG	ATGGCTCTATCGGTGTCAGC

## Data Availability

Data are contained within the article.

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
