# Peer review of "Phenotypic Analysis and Gene Cloning of Rice Floury Endosperm Mutant wcr (White-Core Rice)"

_plants, 2024, doi:10.3390/plants13182653_

Round 1
Reviewer 1 Report
Comments and Suggestions for Authors
In this manuscript, Yang and his/her colleagues identified a floury endosperm mutant wcr. Furthermore, they investigated the agronomic traits and conducted physical and chemical analyses of storage substances, and found that starch and protein components are differentially influenced spatially. Genetic analysis coupled with map-based cloning verified that wcr corresponds to a loss-of-function allele of previously reported OsPPDKB/FLO4. Overall, this manuscript is well organized, but also have some issues that should be addressed for the consideration of its publication in Frontiers in Plant Science.
Major comments:
1. Introduction section needs to be reorganized, especially regarding the advance of PPDKs in plants. In addition, several novel flo mutants have been reported, which should be updated.
2. The authors claimed that plant height and tiller number are also altered in the wcr mutant, which obviously contradicts with previous reports. How to explain this inconsistence?
3. Does PPDK participate in low oxygen stress at the seedling stage? Functional research in this issue can increase the novelty of this manuscript.
4. A major novelty of this manuscript is the differential content of storage substances in different part in the wcr mutant, what is the possible cause of this phenomena? Is it correlated with the spatial expression of the target gene?
Minor comments:
1. The writing of this manuscript requires substantial improvement. Many phrase and sentences are wrongly constructed, which need to be corrected throughout this manuscript.
2. Mutant wcr requires full splicing when it first appears.
3. Line 14: “floury” instead of “powdery”.
4. Line 18: “milled rice” instead of “polished rice”.
5. The authors should check the writing of proteins and genes, especially when they describe mutants (e.g. lines 21, but also other regions of the paper).
6. Line 35: but also other regions of the paper: Grasses have no seeds but fruits, i.e. grains or caryopses. The text is to be corrected accordingly.
7. Figure 1B, C: Lack of scale bar.
8. Figure 1D: The scanning electron microscope image of the wild-type endosperm shows the cell wall without revealing starch granules. Please replace the image.
9. Figure 2H: All abbreviations have to be explained at their first use.
10: Line 163: the authors should provide the genetic information of SL418.
11: Line 168: OsPPDKB or OsPPDKb ? The description should be uniform throughout the entire manuscript.
12: Line 169: An additional base C into one (“the first” in the abstract) exon of the wcr gene. The authors should check it.
13: Figure 4D: Add the sanger chromatogram of wild-type DNA fragments.
14: Line 179: Please add materials and methods of qRT-PCR of 53 genes and give information on the accession number and primer sequence of each gene.
15: Figure 5: It is better to split this figure into two graphs, one representing protein synthesis and the other representing starch metabolism.
Comments on the Quality of English LanguageThe writing of this manuscript requires substantial improvement. Many phrase and sentences are wrongly constructed, which need to be corrected throughout this manuscript.
Author Response
Comments 1: Introduction section needs to be reorganized, especially regarding the advance of PPDKs in plants. In addition, several novel flo mutants have been reported, which should be updated. |
Response 1: Thank you for pointing this out. We have reorganized the introduction section, added the advance of PPDKs and novel flo mutants which have been reported, please check the line 40-66. |
Comments 2: The authors claimed that plant height and tiller number are also altered in the wcr mutant, which obviously contradicts with previous reports. How to explain this inconsistence? |
Response 2: Thank you for pointing this out. We added the explanation in the discussion section, please check the line 273-277.
|
Comments 3: Does PPDK participate in low oxygen stress at the seedling stage? Functional research in this issue can increase the novelty of this manuscript. |
Response 3: Special thanks to the reviewers for their constructive comments. As the main purpose of this study is to clone the wcr gene and analyze its regulatory role in rice storage substances, we have not conducted research on seedling response to low oxygen stress at this stage. However, we plan to carry out relevant experiments in the future and look forward to presenting positive results. |
Comments 4: A major novelty of this manuscript is the differential content of storage substances in different part in the wcr mutant, what is the possible cause of this phenomena? Is it correlated with the spatial expression of the target gene? |
Response 4: Agree. We highly agree with your opinion, and in the discussion section, we focused on exploring possible reasons for this phenomenon. Please check the line 245-272.
|
Comments 5: The writing of this manuscript requires substantial improvement. Many phrase and sentences are wrongly constructed, which need to be corrected throughout this manuscript. |
Response 5: Thank you for pointing this out. We have made changes and improvements to the language all throughout the article.
|
Comments 6: Mutant wcr requires full splicing when it first appears. |
Response 6: Agree. We have corrected this mistake, please check the line 13.
|
Comments 7: Line 14: “floury” instead of “powdery”. |
Response 7: Agree. We have corrected this mistake. |
Comments 8: Line 18: “milled rice” instead of “polished rice”. |
Response 8: Agree. We have corrected this mistake.
|
Comments 9: The authors should check the writing of proteins and genes, especially when they describe mutants (e.g. lines 21, but also other regions of the paper). |
Response 9: Thank you for pointing this out. We have checked the writing of proteins and genes.
|
Comments 10: Line 35: but also other regions of the paper: Grasses have no seeds but fruits, i.e. grains or caryopses. The text is to be corrected accordingly. |
Response 10: Agree. We have corrected this mistake.
|
Comments 11: Figure 1B, C: Lack of scale bar. |
Response 11: Thank you for pointing this out. We have added the scale bar.
|
Comments 12: Figure 1D: The scanning electron microscope image of the wild-type endosperm shows the cell wall without revealing starch granules. Please replace the image. |
Response 12: Thank you for pointing this out. But we are very sorry, we couldn't find any other images that can show the wild-type starch granules at this scale. At this scale, the wild-type starch granules are densely packed, while the mutant ones are very sparse, which better reflects the state of endosperm starch granules between the two materials. |
Comments 13: Figure 2H: All abbreviations have to be explained at their first use. |
Response 13: Thank you for pointing this out. We have explained the abbreviations in the caption, please check the line 129-130.
|
Comments 14: Line 163: the authors should provide the genetic information of SL418. |
Response 14: Thank you for pointing this out. Since this chromosome substitution segment line (SL418) is not constructed by us, we have cited published literature. Please check the line 165.
|
Comments 15: Line 168: OsPPDKB or OsPPDKb ? The description should be uniform throughout the entire manuscript. |
Response 15: Thank you for pointing this out. It should be OsPPDKB. We have uniformed the description throughout the entire manuscript.
|
Comments 16: Line 169: An additional base C into one (“the first” in the abstract) exon of the wcr gene. The authors should check it. |
Response 16: Thank you for pointing this out. We have corrected this mistake, please check the line 171.
|
Comments 17: Figure 4D: Add the sanger chromatogram of wild-type DNA fragments. |
Response 17: Thank you for pointing this out. We have added the sanger chromatogram of wild-type DNA fragments.
|
Comments 18: Line 179: Please add materials and methods of qRT-PCR of 53 genes and give information on the accession number and primer sequence of each gene. |
Response 18: Thank you for pointing this out. Since the primer information for these 53 genes has been reported, we cite the relevant literature. Please check the line 348. |
|
Comments 19: Figure 5: It is better to split this figure into two graphs, one representing protein synthesis and the other representing starch metabolism. Response 19: Thank you for pointing this out. We have split this figure into two graphs. |
|
4. Response to Comments on the Quality of English Language |
Point 1: The writing of this manuscript requires substantial improvement. Many phrase and sentences are wrongly constructed, which need to be corrected throughout this manuscript. |
Response 1: Thank you for pointing this out. We have made changes and improvements to the language all throughout the article. Please check the revised section of the article |

Reviewer 2 Report
Comments and Suggestions for Authors
The main objective of this manuscript is to describe the vcr mutation in rice. The manuscript was medium-done written and had good results. However, there are many English grammar, punctuation, and spelling mistakes. There is a lack of order regarding data interpretation, and some issues must be corrected after acceptance. Throughout the manuscript, I pointed out some questions and suggestions. Pl use these suggestions to create a new manuscript and return it in a new submission or after Major Revision in accordance with the Editor's recommendation.
Pay attention to self-citation. Of the last six cited articles, at least half are self-citations; please check the actual intent.

Comments on the Quality of English LanguageThere are many errors in correct English writing, grammar, and spelling form. The entire manuscript must be sent to a multidisciplinary team for grammatical and formal corrections and the proper explanation of scientific data. I tried to correct some errors, but I am also not a professional suited for this grammatical and literary correction. Here are just a few suggestions.
Round 2
Reviewer 1 Report
Comments and Suggestions for Authors
The authors have been addressed all the concerns I have, and I have no more questions on this MS.
Author Response
Comments 1: The authors have been addressed all the concerns I have, and I have no more questions on this MS.
Response 1: Thank you very much for your efforts and help.
Reviewer 2 Report
Comments and Suggestions for Authors
The authors have diligently adhered to most of the suggested revisions, demonstrating their commitment to improving their work. They have notably enhanced the structure and clarity of the introduction, rectified methodological errors, and restructured the results section. However, some of the reviewer’s concerns (such as the limited scope of the discussion and detailed gene explanations) were only partially addressed. Below more detail:
Introduction:
While the structure and logic improved, the introduction remains slightly superficial in explaining the mutant's importance beyond its phenotypic analysis. A stronger connection to the study’s goals would benefit clarity.
In Materials and Methods, the authors addressed several methodological concerns, including providing better explanations for sample preparation and correcting the protein determination method to the Bradford method. However, they still failed to fully describe some reagents and equipment details essential for reproducibility (e.g., clarification of specific concentrations and technical replicates).
Results:
The figures were reorganized, and data reporting was improved in places. However, some issues, such as mixing results with interpretations and conclusions, persisted. Furthermore, the reviewer’s suggestion to replace horizontal lines in figures with statistical notations (upper and lowercase letters) was not fully implemented. Also, this section needs a more precise reference to figures and tables that should be included in the text, and some interpretations should be moved to the discussion section.
Discussion:
The authors expanded the discussion but still focused heavily on their own results without making enough comparisons to existing research. Thus, the reviewer’s critique of limited referencing was not fully addressed, and it is still lacking in depth. The discussion would benefit from a more balanced critique of the results and greater incorporation of relevant literature to contextualize the findings better.
Comments on the Quality of English Language
The manuscript would benefit from a linguistic revision to improve fluency, particularly addressing sentence structure, verb agreement, and punctuation. These adjustments would make the text smoother and more professional. However, no major rework is needed as the scientific content and vocabulary are accurate.
I recommend a minor linguistic revision to refine grammar, eliminate redundancy, and enhance the manuscript's overall readability.
Some examples:
“Rice is a crucial staple crops worldwide” should be “Rice is a crucial staple crop worldwide”
"There is no difference in the protein content of the other three components when analyzing milled rice flour" could be clearer if it were changed to "There is no difference in the protein content of the other three components when analyzed in milled rice flour."
In several sections, phrases like "It can also be inferred" and "based on these results" appear repetitively without adding significant value. Reducing redundancy will streamline the text.
